# Wearable Sensor for Forearm Motion Detection Using a Carbon-Based Conductive Layer-Polymer Composite Film

**DOI:** 10.3390/s22062236

**Published:** 2022-03-14

**Authors:** Kiwon Park

**Affiliations:** Department of Mechanical & Automotive Engineering, Youngsan University, Junam-ro 288, Yangsan-si 48015, Korea; kwp@ysu.ac.kr

**Keywords:** carbon-based conductive layer-polymer composite film, motion detection algorithm, sensor fabrication, wearable sensor

## Abstract

In this study, we developed a fabrication method for a bracelet-type wearable sensor to detect four motions of the forearm by using a carbon-based conductive layer-polymer composite film. The integral material used for the composite film is a polyethylene terephthalate polymer film with a conductive layer composed of a carbon paste. It is capable of detecting the resistance variations corresponding to the flexion changes of the surface of the body due to muscle contraction and relaxation. To effectively detect the surface resistance variations of the film, a small sensor module composed of mechanical parts mounted on the film was designed and fabricated. A subject wore the bracelet sensor, consisting of three such sensor modules, on their forearm. The surface resistance of the film varied corresponding to the flexion change of the contact area between the forearm and the sensor modules. The surface resistance variations of the film were converted to voltage signals and used for motion detection. The results demonstrate that the thin bracelet-type wearable sensor, which is comfortable to wear and easily applicable, successfully detected each motion with high accuracy.

## 1. Introduction

Owing to the ability of wearable devices to collect and transmit both internal and external physiological information in real time, they have applicability in various fields, such as security [1], medicine [2,3], welfare of the elderly [4,5], device control [6], health monitoring [7,8], and entertainment [9].

Wearable devices essentially involve the detection of human movement intention [1], most commonly through electromyography (EMG) signals from the body [9,10,11,12,13,14,15]. An EMG signal is an electrical stimulus transmitted from the brain to the muscle fiber to contract the muscle. Methods of signal measurement include invasive methods, wherein a needle-shaped electrode is inserted into the peripheral nervous system of the body, and non-invasive methods [12,14], wherein a different type of electrode is attached to the surface of the skin to measure the surface EMG (sEMG) signal [9,11,13,15]. Generally, non-invasive methods that can quantitatively analyze the overall synergistic activities of a set of muscular movement units are used [16]. However, given that sEMG signal measurement requires a technique to extract the motion intention from complex muscle noise signals generated from movements inside and outside the body, many studies have applied fuzzy systems and neural networks that showed excellent effects on nonlinear signal processing [13,16,17]. Additionally, because the characteristics of signals vary depending on the attachment position of the electrode and muscle fatigue, it is difficult to ensure reproducibility when EMG signals are used as control signals for prosthetics in long-term applications [10,16]. Moreover, multiple electrodes need to be attached to the surface of the body: one to the muscle group and another to ground for differential amplification [13,16,17,18]. Given the inconvenience related to the utilization of sEMGs, researchers have developed a sensor system using commercialized sensors, such as accelerometers, strain gauges, and tactile sensors, to detect motion intention in the body without using EMG signals. Furthermore, research has been conducted on developing sensors suitable for detecting human movement.

Wearable devices that utilize accelerometers to detect position and movement have been applied in fields such as sports training and physical therapy [1,19,20,21]. Furthermore, accelerometers have been applied to inertial navigation systems to monitor the position, direction, and speed of moving objects to detect human movement and track the position of individuals indoors [22,23,24,25].

Wearable devices using strain-gauge sensors detect movement intention based on the change in resistance owing to the contraction and expansion of the skin surface caused by movement. In a prior study, multiple strain-gauge sensors were attached to a wearable upper-arm band fabricated from silicone to detect the movements of the upper arm [18]. In addition, a strain-gauge element with a specific pattern was fabricated and attached to a flexible film, which, in turn, was attached to the body to measure strain and detect movement [26,27,28]. Researchers have also used liquid-phase metals [29,30,31,32,33] and force sensors to develop elastic strain-gauge sensors for the detection of expansion pressure produced during muscle contraction to track movement intention [34,35,36].

A human movement intention study was conducted using a tactile sensor to detect flexion changes on the skin surface [37,38,39,40]. Furthermore, researchers have developed a wearable upper-arm sensor comprising multiple tactile sensors to detect movement [37,38,39,40]. In addition, a study was conducted using a thimble-shaped tactile sensor that was wearable on the finger to detect its movement intention [41].

Regarding signal processing algorithms that recognize signals generated from complex structures in the human body, methods showing excellent performance in nonlinear and nonspecific data processing, such as fuzzy logic and neural network theory, are being extensively used [11,12,13,30,42,43,44,45]. Additionally, prior studies have combined fuzzy logic and neural network theory to complement each other [46,47,48]. However, learning based on repetition must be conducted beforehand to adjust the connection strength within the internal network [49]. Generally, as the number of learning iterations increases, the error between the command signal and actual movement decreases. Moreover, to reduce the vast amount of EMG signal data measured in real time, a fuzzy c-means clustering technique was used to group similar data and utilize only the center values [47,50].

Recently, there has been an increase in the use of carbon in the fabrication of strain-gauge-type sensors, and most applications have used carbon nanotubes (CNTs) as a key material. Research on the application of carbon paste on polymer film to develop a strain-gauge-type wearable sensor has not yet been reported. The use of CNTs helped improve sensitivity and enabled large surface measurement, which is ideal for applications in wearable devices [51]. However, practical applications have not yet been reported. CNT-coated thread was used as a wearable sensor to record the mechanomyography of leg muscles, and it showed promising results for assessing muscular activity [52]. A similar type of CNT-coated thread-based fabric was developed as a pressure sensor [53]. A CNT-based formulated ink printed in a pattern on the textile surface was used as a wearable temperature sensor [54], and a CNT-coated polyacrylonitrile film based on the electrospinning method was applied to respiration monitoring [55].

In this study, a method to fabricate a bracelet-type wearable sensor module using a carbon-based conductive layer-polymer composite film was developed to detect the movement intention of the human body. A polyethylene terephthalate (PET) polymer film with a conductive layer made using a carbon paste was used as the integral material for the composite film to detect changes in resistance corresponding to the flexion changes of the surface of the body owing to the contraction and relaxation of muscles. The mechanism of surface resistance variation of the composite film employed in this study was previously investigated by K. Park et al. [56], who identified a relationship between the variations that occur in surface resistance and the width of crack gaps in the conductive layer when the film is bent. Their results showed that the width of the cracks significantly affected the resistance variation of the conductive layer.

Variations in surface resistance were converted into voltage signals and used to detect the movement intention of the forearm. A recognition algorithm was also designed, and we verified its performance in detecting wrist flexion and extension, ulnar deviation, and finger flexion movements based on the voltage signals measured using the proposed bracelet sensor attached to the upper arm, as shown in Figure 1. Because the wearable sensor produced voltage signals corresponding to flexion variation, the signal could be easily interpreted compared to a complex EMG signal, and detection of movement intention was possible using a simple algorithm rather than complex pattern recognition algorithms, such as fuzzy logic and neural network algorithms. In addition, the proposed sensor technique integrates the position of the sensor to where the bracelet is worn, thereby reducing the inconvenience of large surface requirements and the number of sensors compared to the EMG technique, which requires multiple sensor attachments all around the upper arm to detect the same motions.

## 2. Sensor Mechanism and Fabrication Method

### 2.1. Fabrication of the Carbon-Based Conductive Layer-Polymer Composite Film

In this study, we fabricated a composite material film with a conductive layer, the resistance of which changed with the degree of flexion, to detect flexion change during body movement. When a strip-shaped film, the surface of which is coated with a conductive layer, is bent in the longitudinal direction, cracks form in the conductive layer, which changes the resistance. However, different forces that depend on the flexion direction affect the size of the cracks and the degree of damage to the surface electrodes [56].

When the conductive layer is located on the outer surface of the film, fine cracks with a thickness less than 20 μm are formed because of the tensile force during the flexion of the film, which increases the surface resistance [56]. However, when the film returns to the equilibrium state, the cracks close, which leads to a decrease in the surface resistance, as shown in Figure 2a.

In contrast, when the conductive layer is located on the inner surface of the film, the cracks generated owing to compressive force are much larger compared to those formed because of tensile force, resulting in significant damage to the electrode. As a result, the conductive layer is significantly damaged, and the surface resistance increases as the film returns to the equilibrium state, as shown in Figure 2b.

Therefore, in this study, the conductive layer formed on the surface was positioned on the outer side of the flexion (Figure 2a). The damage to the conductive layer due to the compressive force (Figure 2b) was restricted, and the change in resistance due to flexion (Figure 2a) was used to fabricate the bracelet sensor.

Graphite paste was used to form a conductive layer on the surface of a 0.1 mm thick PET film to improve the durability of the wearable bracelet sensor. The bonding strength between the conductive layer and the PET film was improved by applying a thermal compression method using a press roller with a temperature of 130 °C. Figure 3 illustrates the fabrication procedure of the carbon-based conductive layer-polymer composite film. Figure 4 shows a comparison of the durability of the conductive layer with and without the heat-press procedure. It can be seen that the conductive layer formed without the heat-press procedure was significantly damaged after repeated bending; however, the conductive layer formed using the heat process exhibited higher durability, as shown in Figure 4c.

Figure 5 shows the scanning electron microscopy (SEM) image of the cracks formed in the conductive layer when exposed to compressive and tensile forces. The sample film was sputtered by Pt, and SEM images were obtained using SU8220 cold FE-SEM to capture secondary electron images under an acceleration voltage of 5 kV. Although the conductive layer was plated using the mechanical adhesive method shown in Figure 3, which differs from the electrochemical plating method in [56], the cracks formed owing to compressive force were more significant than those formed because of tensile force, as shown in Figure 5b,c, which supports the crack-generation mechanism in Figure 2. Furthermore, the cracks formed under compressive force are considerably wider than those formed under tensile force in both the vertical and horizontal directions after the material returns to the flat position.

### 2.2. Fabrication of the Proposed Bracelet Sensor

When a 0.1 mm thick composite material film is used directly on the body, the conductive layer can easily get damaged, owing to external physical effects, such as skin friction, potentially generating static electricity, which may cause the film to fold in the wrong direction. This results in exposure to a compression force that makes reproducibility of the sensor signal difficult. Therefore, to minimize such external physical effects, a sensor module was designed, fabricated, and implemented for data collection and motion detection algorithms.

As shown in Figure 6, the final sensor module comprises two mechanical parts. The first part is inserted inside the other part and connected using a flexible band (band A, Figure 6a), which causes sliding movement corresponding to the contraction and expansion of the skin surface. The resulting flexion of the composite film connected to both mechanical parts maintains the curvature so that the conductive layer of the film is always exposed to tensile stress, as shown in Figure 2a, to prevent damage to the surface electrode, as shown in Figure 2b. The conductive film was cut into smaller parts with dimensions of 2 (w) × 18 (l) (mm) that were placed on each module. The total size of the assembled module was 12.0 (w) × 30.5 (l) × 5.0 mm (h). The length of the gap between each mechanical part (x (mm) in Figure 6b) can be stretched from 0 to 3 mm, and the width of the flexible band connecting each module is 9 mm.

The fabricated bracelet-type sensor consisted of three sensor modules. Figure 7 shows the variation of resistance corresponding to the stretched length of each module, which is represented by x (mm) in Figure 6b. The resistance variations shown in Figure 7 indicate that each fabricated sensor module has enough sensitivity to detect even smaller variations of flexion on the human body. The fabricated bracelet sensor was then worn such that the three sensor modules were located around the surface of the forearm muscle groups shown in Figure 8: extensor carpi ulnaris, flexor carpi ulnaris, and flexor digitorium superficialis.

Therefore, when the muscles contract and the skin protrudes, the sensor module stretches in the longitudinal direction, and the resistance of the conductive layer decreases, as shown in Figure 6c. Conversely, when the muscles expand and the skin surface becomes depressed, the sensor module contracts in the longitudinal direction, and the resistance of the conductive layer increases, as shown in Figure 6c.

Compared to the band connecting the sensor modules (band B, Figure 6a), the band connecting the two mechanical parts in the sensor module (band A, Figure 6a) exhibits low elasticity. Therefore, protrusions and depressions can be effectively detected by the sensor module, as shown in Figure 6c.

## 3. Signal Measurement

The conventional EMG method used to detect movement of the body requires an electrode on each muscle group [18]. Therefore, it requires a total of nine electrodes, which includes one ground electrode on the anatomical locations of the muscle groups, as shown in Figure 8, to detect the four motions of the forearm: wrist extension, ulnar deviation, wrist flexion, and finger flexion. However, in this study, we used only three sensor modules comprising a bracelet sensor developed to measure the signals generated by the four motions.

Figure 9 shows the transducer circuit comprising a bridge circuit and a differential amplifier IC (INA114) for each module to convert surface resistance variations into voltage signals. A 50-KΩ R_G_ was used to obtain amplification by a factor of 50. The signals transmitted by each sensor module on the upper arm were collected using the NI-DAQ and analyzed using MATLAB simulator at a sampling frequency of 1 kHz.

The signal was measured for 15 s. For the first 5 s for each movement, the forearm remained rested on the table without any movement. Then, each movement was performed for approximately 5 s, followed by a resting state for 5 s. All movements were performed at a constant speed.

## 4. Design of the Motion Detection Algorithm

Figure 10 shows the measured output voltage from the three sensor modules during wrist extension, ulnar deviation, finger flexion, and wrist flexion. To verify the performance of the sensor, a raw signal without a filter was measured and used in motion analysis.

Although the variance of the signal was low in all motions during the resting stage, it was comparatively higher in the beginning and finishing states of each motion. In addition, the magnitude of the voltage showed a discriminative level while maintaining each motion compared to the resting state. Therefore, the resting and active states were accurately detected.

Figure 10a shows the voltage waveforms generated during the resting state and wrist extension motion. During wrist extension, the muscles touching the sensor modules contract and the skin surface protrudes. As a result, the sensor modules stretch in the longitudinal direction, and the resistance of the conductive layer decreases, thereby decreasing all signals. Conversely, during ulnar deviation, finger flexion, and wrist flexion motions, the muscles touching the sensor module expand, and the skin surface becomes depressed. As a result, the sensor module contracts in the longitudinal direction, and the resistance of the conductive layer increases, thereby increasing all signals, as shown in Figure 10b–d. It was observed that all the motions produced voltage signals at different levels. In wrist extension, the magnitudes of signal variance from sensor modules 2 and 3 were the highest and lowest, respectively. Similarly, during ulnar deviation, the magnitudes of signal variance from sensor modules 1 and 3 were the highest and lowest, respectively. During finger flexion, the magnitudes of signal variance from sensor modules 2 and 3 were the highest and lowest, respectively. Lastly, during wrist flexion, the magnitudes of signal variance from sensor modules 1 and 3 were the highest and lowest, respectively.

For real-time applications, an algorithm was designed to detect each motion based on waveform analysis. Furthermore, to reduce the number of sensors, in contrast with the conventional EMG method, detection algorithms using signals from two and three sensor modules were designed, and the corresponding performances were compared.

Figure 11 shows the algorithms used to detect four motions using two and three sensor modules, respectively, where S_1_, S_2_, and S_3_ are the voltage signal data from sensor modules 1, 2, and 3, respectively. O_1_, O_2_, and O_3_ are the means of the data gathered during the resting state from sensor modules 1, 2, and 3, respectively. M_12_ is expressed in Equation (1), and |M_12_| is the absolute value of M_12_.
M_12_ = (S_1_ − O_1_) − (S_2_ − O_2_)(1)
where C_1_, …, C_17_ are the threshold values listed in Table 1, which were selected as the boundary values of M_12_ and |M_12_| in each motion for improved detection accuracy through the simulation process.

The motion detection algorithms shown in Figure 11 were implemented in the MATLAB simulator and were used to detect four motions of the forearm. As the sampling frequency of the signal measurement was 1 kHz, 15,000 points of data were collected during the measurement time (15 s). The measured data from the three sensor modules were loaded into MATLAB, and the mean value for every five data points, i.e., the raw voltage signals (S_1,2,3_) in Figure 11, was sequentially inputted to the algorithm.

## 5. Results and Discussion

Figure 12 shows the detected motions of the forearm using the measured signals, where figures ‘A’ and ‘B’ are the results using two and three sensor modules with the algorithms in Figure 11a and Figure 11b, respectively. The points on the graph that express the intention of motion are as follows: 1: rest state; 2: wrist extension; 3: ulnar deviation; 4: finger flexion; and 5: wrist flexion. Figure 12a shows the classification results for the wrist extension motion, which was performed from approximately 4 to 9 s; therefore, ‘2’ is classified correctly. For the other intervals, because the forearm of the test subject was in the resting state, ‘1’ is classified correctly. Figure 12b shows the classification results for the ulnar deviation motion, which was performed from approximately 5 to 9 s. Although ‘3’ is mostly correctly classified, as shown in figure ‘A’ in Figure 12b, the wrist flexion motion outputs were observed at the beginning and end of the ulnar deviation motion, owing to the limitation of the two-sensor module algorithm. In other words, the boundary conditions in the ‘if’ statement for detecting ulnar deviation overlapped with those of wrist flexion in the algorithm shown in Figure 11a. However, this issue was resolved when three sensor modules were used, as shown in Figure 12b, and the finger flexion and wrist flexion motions were successfully detected with no observed differences between the two- and three sensor modules, as shown in Figure 12c,d.

Figure 12 shows a successful detection of each motion without error by using the thin bracelet-type wearable sensor composed of three sensor modules; the sensor was comfortable to wear and easy to apply. From the comparison of the results of this study with those of other studies that used CNTs [51,52,53,54,55], it was observed the use of conductive carbon paste is more cost-effective and is easy to handle compared to CNTs. However, to guarantee the durability and reproductivity of soft-film-type sensor material, the design and fabrication of a mechanical module to guide it is required, and a minimization technique of guider to realize comfortable wearable sensor is necessary. In addition, the result verifies the resistance variation mechanism caused by the cracks on the conductive layer by bending of film, previously studied in [56]. Moreover, the tiny mechanical guide designed to fit wearable sensor system seamlessly detects the flexion variation of the body surface. Because the simple bending motion of the polymer film produces a signal with high sensitivity, the proposed method has promising applications in wearable sensor technology to detect body movement. It can also be widely applied in various types of mechanical guides.

## 6. Conclusions

In this study, we developed a method for fabricating a variable-resistance bracelet sensor using a flexible polymer film-conductive layer composite film to detect changes in flexion motion due to muscle contraction and relaxation. In addition, a motion detection algorithm was devised to detect four movements—wrist flexion, wrist extension, ulnar deviation, and finger flexion—based on the signals measured using two- and three-sensor modules in the proposed sensor when attached to the forearm, and its performance was verified. Finally, the motion detection performances of the two- and three-sensor modules were compared and analyzed.

Results showed that the motion detection algorithm successfully detected the motions of the forearm using the developed bracelet sensor without using complex classification theories. Furthermore, the error that occurred when detecting ulnar deviation motion using the two-sensor module was eliminated successfully using the three-sensor module. Moreover, compared with the conventional EMG technique, the number of sensors in contact with the skin surface was reduced by more than 70%, and the patch sensors used in the EMG method that would cover most parts of the forearm were substituted with a single, thin bracelet-type sensor that can be worn around the forearm. As a result, the proposed device should be comfortable to wear and easy for practical applications.

## Figures and Tables

**Figure 1 sensors-22-02236-f001:**
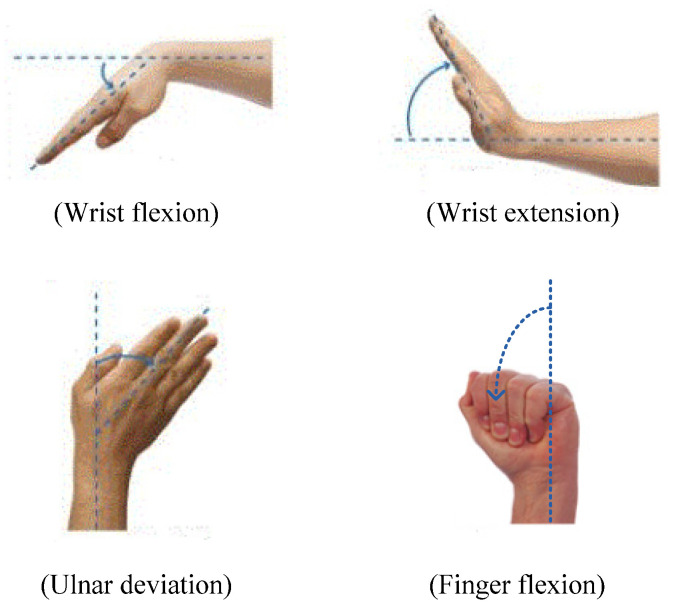
Forearm movements.

**Figure 2 sensors-22-02236-f002:**
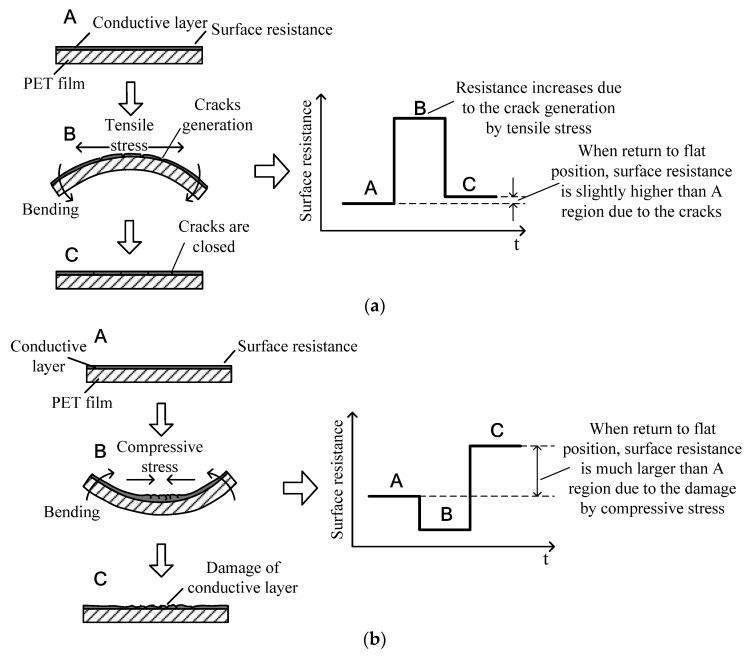
Changes in the resistance of the conductive layer owing to the crack created in the conductive layer by tensile and compressive forces depending on the flexion direction (A: flat state, B: bent state, and C: returned state from bending). (**a**) Degree of damage and change in resistance of the conductive layer owing to tensile force; (**b**) degree of damage and change in resistance of the conductive layer owing to compressive force adapted with permission from ref. [56]. 2022 IOP Publishing. (K. Park et al., Smart material and structures, vol. 19, 2010, IOP Publishing).

**Figure 3 sensors-22-02236-f003:**
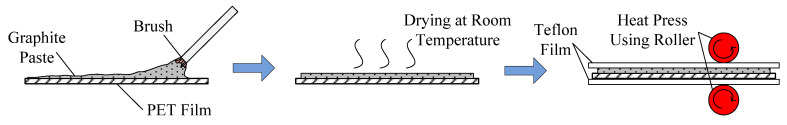
PET film—graphite-paste composite film fabrication.

**Figure 4 sensors-22-02236-f004:**
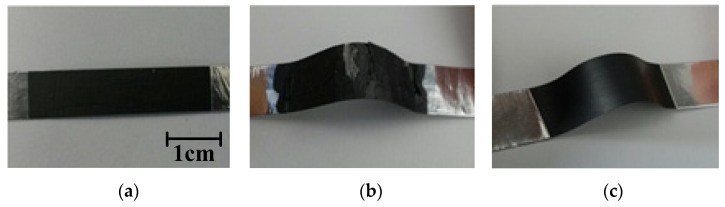
Durability of the composite film according to the fabrication process. (**a**) Form of the fabricated composite film; (**b**) only the drying process is applied (conductive layer is damaged from repeated flexion); (**c**) both drying and thermal compression are applied (improved durability without damaging the conductive layer after repeated flexion).

**Figure 5 sensors-22-02236-f005:**
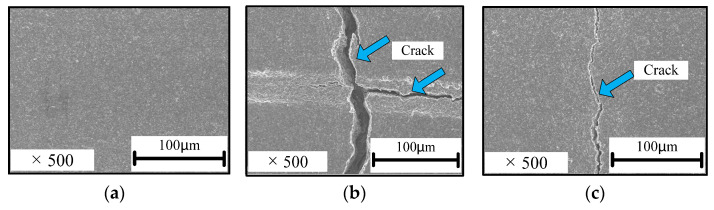
Scanning electron microscopy (SEM) images of the cracks formed in the conductive layer owing to compressive and tensile forces. (**a**) Equilibrium before flexion; (**b**) cracks formed because of compressive force; and (**c**) cracks formed because of tensile force.

**Figure 6 sensors-22-02236-f006:**
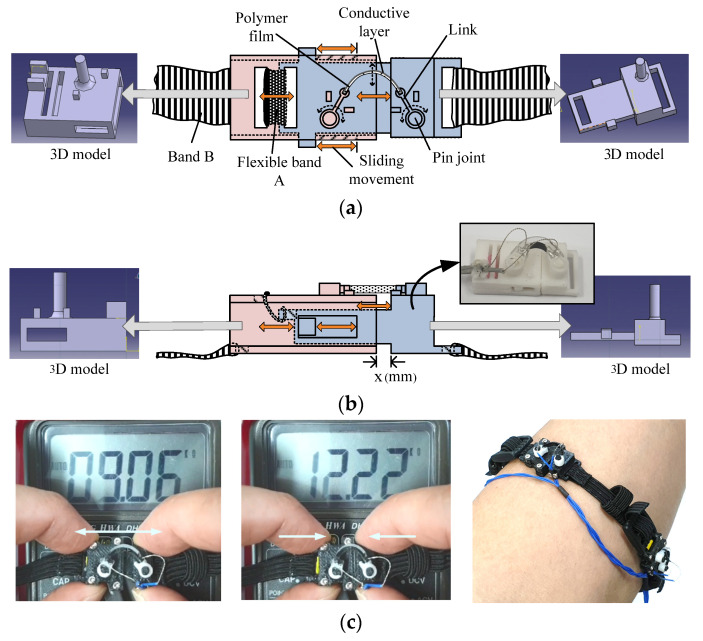
Movement mechanism and fabricated form of the sensor module. (**a**) Top view; (**b**) side view with fabricated image of the module; (**c**) voltage signal variation due to the surface resistance change of the composite film mounted on the sensor module; image of the final version of the sensor module.

**Figure 7 sensors-22-02236-f007:**
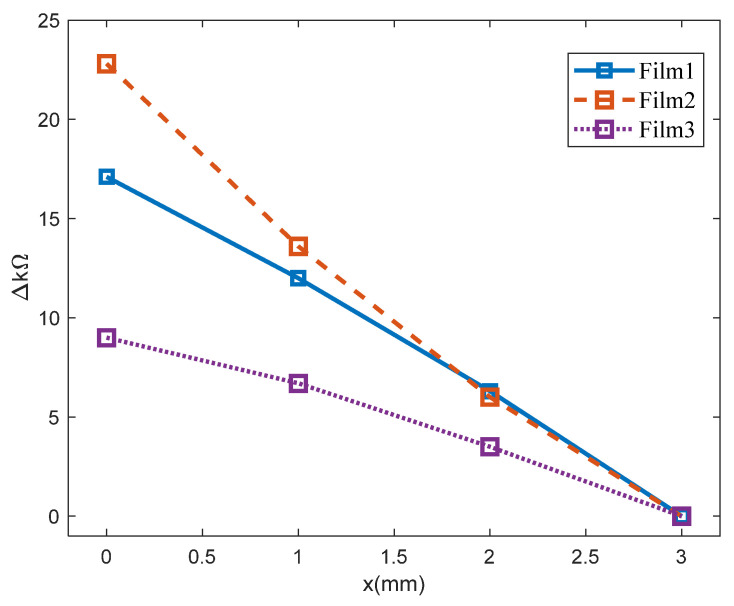
Surface resistance variations on the three films in each sensor module.

**Figure 8 sensors-22-02236-f008:**
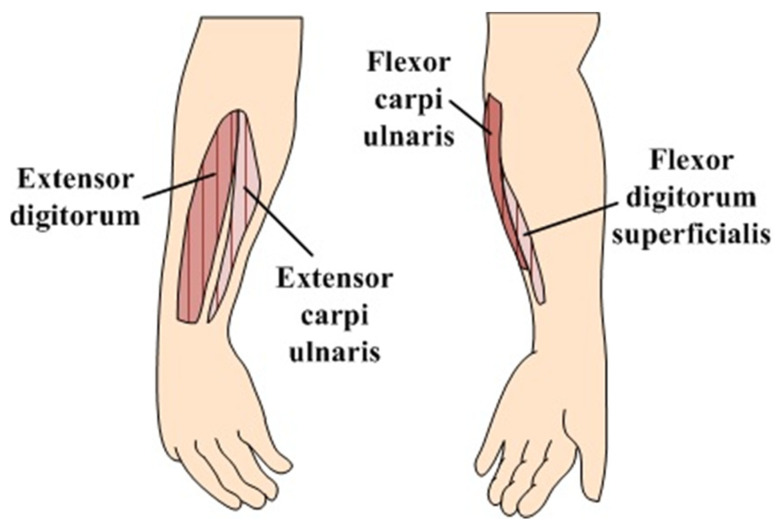
Anatomical locations of the forearm producing four motions: wrist extension, ulnar deviation, wrist flexion, and finger flexion.

**Figure 9 sensors-22-02236-f009:**
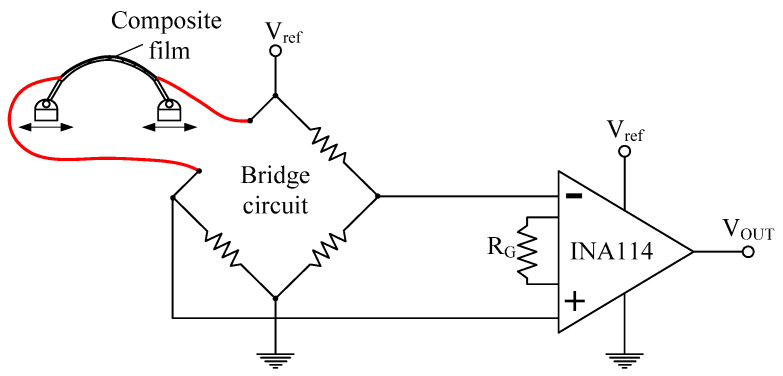
Transducer circuit design for converting resistance variation into voltage signals.

**Figure 10 sensors-22-02236-f010:**
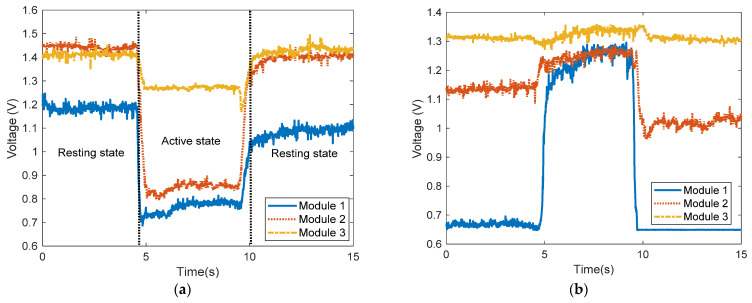
Measured signals from forearm motions. (**a**) Wrist extension, (**b**) ulnar deviation, (**c**) finger flexion, (**d**) wrist flexion.

**Figure 11 sensors-22-02236-f011:**
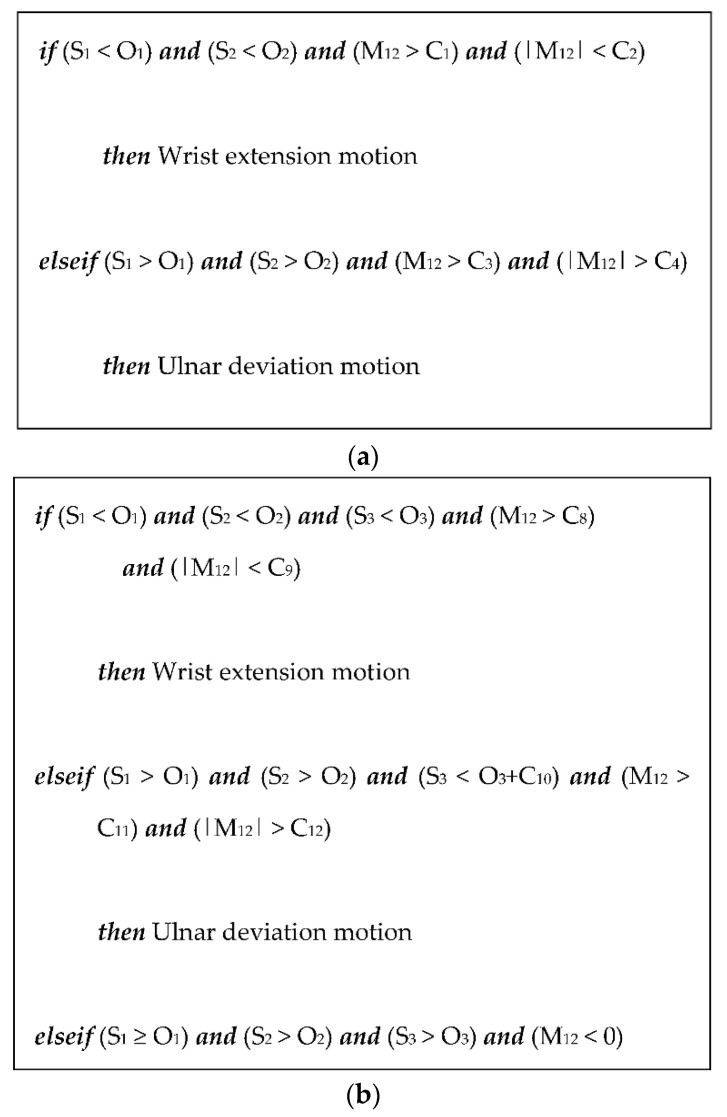
Algorithms to detect forearm motions. (**a**) Algorithms using signals from two-sensor modules and (**b**) algorithms using signals from three sensor modules.

**Figure 12 sensors-22-02236-f012:**
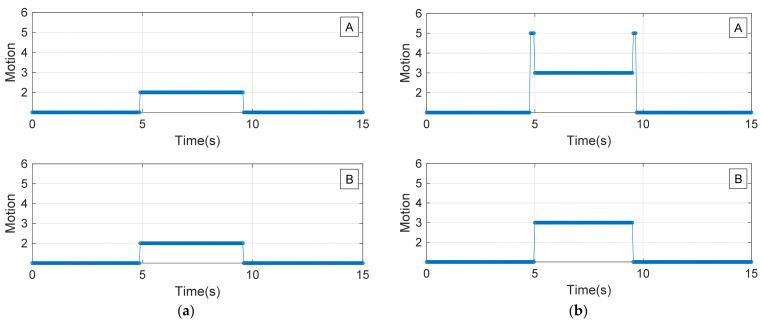
Motion detection results (A: result using two sensor modules and B: result using three sensor modules). (**a**) Wrist extension, (**b**) ulnar deviation, (**c**) finger flexion, and (**d**) wrist flexion.

**Table 1 sensors-22-02236-t001:** Threshold values.

	**C_1_**	**C_2_**	**C_3_**	**C_4_**	**C_5_**	**C_6_**	**C_7_**	**C_8_**	**C_9_**	
Value	0.10	0.20	0.30	0.30	0.53	−0.12	0.50	0.10	0.20	
	**C_10_**	**C_11_**	**C_12_**	**C_13_**	**C_14_**	**C_15_**	**C_16_**	**C_17_**		

Value	0.10	0.30	0.30	0.53	0.03	−0.40	0.40	0.40		


## Data Availability

Not applicable.

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
