# Peer review of "Wearable Sensor for Forearm Motion Detection Using a Carbon-Based Conductive Layer-Polymer Composite Film"

_sensors, 2022, doi:10.3390/s22062236_

Round 1

Reviewer 1 Report

Park discusses a wearable sensor solution for forearm motion detection. The topic is hot in the current R&D of human-machine interfacing, medical & wellbeing as well as robotics. The study is comprehensive, however, some aspects of the writing, especially the discussion, need improvements. Either authorship, acknowledgements, or Conflict-of-Interest statement or any combination thereof require a revision.

#01. page 1 line 27: for an interdiciplinary audience, clustering of ref.s is not helpful. Please consider particular assignment to items, i.e. security [REF], medicine [REF], welfare ... [REF] etc.

#02. page 2 lines 81 ff: In this paragraph, I would expect a brief overview about different approaches for carbon-based wearable sensor modules, check literature for carbon-based mechanical sensor applications (e.g. carbon nanotubes) and shortly describe and qualify/quantify them here.  Consequently, in the discussion part (page 10 lines 314 ff), evaluate your method with selected methods from this literature study.

#03. Page 4 line 122: captions Fig. 2: Please adjust the position or role of Ref. [52] as it does not seem to relate to a scientific statement. If it is related to the Figure panels themselves, note that publishers may invoke special Copyright phrases such "Figure adapted to Author et al., Journal, Volume, ID, Year, Publisher (in clear text). Please check that.

#04. Page 4 line 123: physical quantity (<10 µm) remains unexplained, please clarify (crack width? inner diameter?)

#05. Page 4 lines 147 ff: Figure 4

  • A. please introduce scalebar in at least one of the pictures (at least verbally in the caption)
  • B. please put panels to identical dimensions keeping aspect ratio (panel c is unnecessarily higher
  • C. please consider a slight enlargement in order to use page width (you could omit writing labels (a)-(c) below and integrate them into the pictures

#06. Page 4 line 156: once a reference acts as a clear grammatical item, usage of "Ref. [52] (instead of just "[52]") is encouraged.

#07. Page 4 lines 153 ff: please include technical details of the SEM measurements, e.g. filter (SE2 or InLens) and electron energy. Please also state if the surface was sputtered (Pt or C layer, thickness ... ) or not.

#08. Page 5 lines 162 ff: cf. comments #05. B. and C.

#09. Page 6 lines 217 ff: Figure 8

  • Please explain INA114 and Rg in the captions

#10. Page 7 lines 234 ff: Figure 9

  • Please explain "Model 1", "Model 2", "Model 3" in the caption

#11. Page 10 lines 314 ff: Please enhance the discussion with respect to State of the Art as indicated in comment #02.

#12. Page 11 line 340: Acknowledgements.

  • As nowadays it is quite unusal in interdisciplinary engineering research that a full research paper is entirely concepted and written by a single author, absence of acknowledgements seems unreal. From my experience with the matter, I assume that at leaste considerable technical help was exerted by technical staff, that parts of the comprehensive studies might have been performed by students. Please feel free to reconsider authorship of other parties and/or at least add significant sources of assistance in the acknowledgements. If there is really nothing to mention, please comment thereon.
  • Moreover, absence of a funding statement AND declaration of no CoI does not work for me. If there is neither an internal program (maybe Youngsan University internal funding?) nor third-party funding, the work must have been funded by something like the author's personal funds and hence I expect the scientific outcome potentially biased. Please add a clear funding statement or comment thereon.

#

Reviewer 2 Report

The author presents a wearable strain sensor for real-time forearm motion detection. The sensor is made from patterning graphite ink on a PET substrate. The author presents four different forearm motions detection by using three sensing modules with a simple algorithm. Overall, the work is not novel based on the following three reasons: 

1). The strain sensing mechanism (crack-induced resistance change) has been well established. Compared to other sensors' performance, the presented device didn't show any clear advantages.

2). The fabrication methods, the algorithm, and the final demonstration, although simple, are not novel.

3). Key data are missing, for example, strain calibration curves, sensing performance repeatability/reproducibility. 

Therefore, the reviewer cannot recommend the publication of this manuscript.

Reviewer 3 Report

This paper presents a method for fabricating a variable-resistance bracelet sensor using a flexible polymer film-conductive layer composite film to detect changes in flexion motion due to muscle contraction and relaxation. Moreover, a motion detection algorithm was proposed to detect four movements (wrist flexion, wrist extension, ulnar deviation, and finger flexion), based on the signals measured using two and three sensor modules in the proposed sensor when attached to the forearm, and its performance was verified. The motion detection performances of the two and three sensor modules were compared and analyzed.

Shortcoming and missing of the paper are the following:

  1. Text into Figs. 2, 3 is very small.
  2. Figs. 5b and 5c are the same.
  3. What is the relation of qualitative results of Fig. 4 (obvious results for damage and electric resistance behaviors) and Fig. 5 to subsequent quantitative results of the paper?
  4. Fig. 4 is very fine with poorly distinguishable details.
  5. What are the reasons for selection of given values of constants C1C17 in Table 1? If you take other values for these constants, you obtain different results for the paper, or not?
  6. The description of the algorithms in Fig. 10 is insufficient. Why do you select the pointed conditions for the considered motions?

Round 2

Reviewer 2 Report

The manuscript has been improved significantly after the revision. The reviewer recommends the publication of this manuscript in its current form.

Reviewer 3 Report

OK, the author has made necessary revision